# A Cognitive Model for Learning Abstract Relational Structures from Memory-based Decision-Making Tasks

**Haruo Hosoya**[1,2]
[1] Department of Dynamic Brain Imaging, ATR International, Japan
[2] WebLab, Graduate School of Engineering, The University of Tokyo, Japan
`hosoya@atr.jp`

## Abstract

Motivated by a recent neuroscientific hypothesis, some theoretical studies have accounted for neural cognitive maps in the rodent hippocampal formation as a representation of the general relational structure across task environments. However, despite their remarkable results, it is unclear whether their account can be extended to more general settings beyond spatial random-walk tasks in 2D environments. To address this question, we construct a novel cognitive model that performs memory-based relational decision-making tasks, inspired by previous human studies, for learning abstract structures in non-spatial relations. Building on previous approaches of modular architecture, we develop a learning algorithm that performs reward-guided search for representation of abstract relations, while dynamically maintaining their binding to concrete entities using our specific memory mechanism enabling content replacement. Our experiments show (i) the capability of our model to capture relational structures that can generalize over new domains with unseen entities, (ii) the difficulty of our task that leads previous models, including Neural Turing Machine and vanilla Transformer, to complete failure, and (iii) the similarity of performance and internal representations of our model to recent human behavioral and fMRI experimental data in the human hippocampal formation.

## 1 Introduction

In everyday human cognition, we often find relationships among entities. Sometimes, we discern common relational structures across various domains and lift it to general knowledge (Figure 1ab). For example, ordering can naturally be found not only among numbers, but also among objects, among individuals, etc. Thus, after repeated experience of such finding, we spontaneously come up with the abstract notion of ordering independent of concrete entities, which can be key to quick understanding of ordering in unknown domains. This is just one example of *abstract relational structure*; other examples include tree-like structure and cyclic structure.

In recent neuroscience, it has been hypothesized that the hippocampal formation[1] may play a central role in abstract relational representation (Eichenbaum and Cohen, 2014). Indeed, although relational representations have been long known in this neural subsystem (Bunsey and Eichenbaum, 1996; Dusek and Eichenbaum, 1997; Constantinescu et al., 2016; Bao et al., 2019; Park et al., 2020; 2021), newer evidence suggests that such representations can be abstract, i.e., independent of concrete entities (Kumaran et al., 2012; Garvert et al., 2017; Liu et al., 2019). On the theoretical side, Whittington et al. (2018; 2020) have remarkably shown, in their proposal of Tolman-Eichenbaum Machine (TEM), that place- and grid-cell properties in the rodent hippocampus and entorhinal cortex (Moser et al., 2008) can emerge as a result of learning the general relational structure of two-dimensional geometry during spatial random-walk tasks. Thus, abstract relational learning is becoming a fascinating, unified view of the hippocampal computation, potentially explaining previously considered multiple functions such as episodic memory and navigation by a single principle.

---

[1]The hippocampal formation refers to the hippocampus plus its neighbor, the entorhinal cortex.

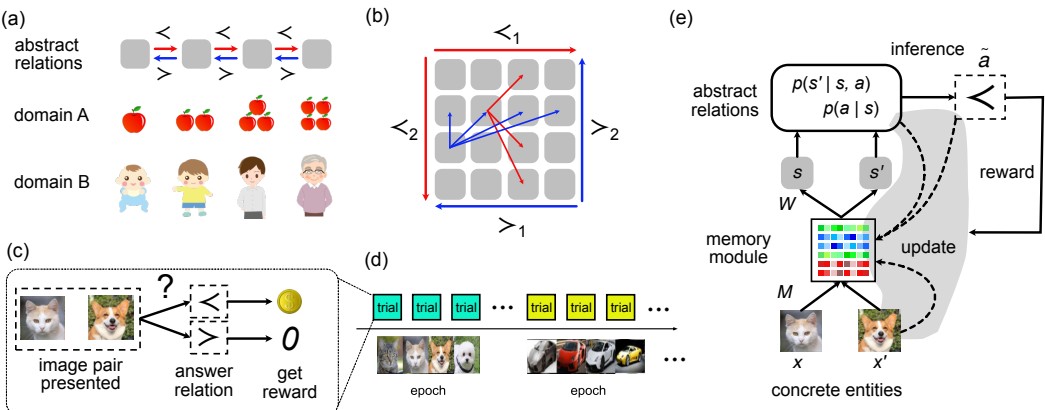

Figure 1: Learning of abstract relational structure. (a) One-dimensional hierarchy (order structure) and its two instantiations for different domains. (b) Two-dimensional hierarchy, with many-to-many relations. (c,d) Relational decision-making task (trial and epochs). (e) Architecture of our proposed model (ARDMO) with relational and memory modules.

In this study, we push this direction further by proposing a novel cognitive model that performs relational decision-making tasks for learning abstract relational structures. Our tasks, inspired by previous human experiments (Kumaran et al., 2012; Park et al., 2020; 2021), require to repeat pairwise relational inference in one domain after another and eventually learn the common relational structure across domains (Figure 1cd). To solve the tasks, building on previous approaches of modular architectures for relational representation (Whittington et al., 2018; 2020), we develop a memory-based learning algorithm that performs reward-guided search for disentangled representations of domain-generic relations and domain-specific entities (Figure 1e). Importantly, we observe a task nature that requires careful choice of a memory mechanism, a key technique to maintain dynamic binding between relations and entities, which leads us to a specific memory update rule that enables content replacement. We call our proposed model Abstract Relational Decision-making MOdel (ARDMO).

Our experiments yielded the following novel findings. First, our model trained on two types of hierarchical relation successfully learned representations that are generalizable to new domains with completely unknown entities and usable in down-stream tasks such as transitive inference. Second, several existing models including Neural Turing Machine (NTM) (Graves et al., 2014), Differentiable Neural Computer (DNC) (Graves et al., 2016), vanilla Transformer (Vaswani et al., 2017), and a version of TEM (Whittington et al., 2018; 2020) could not solve our tasks despite their powerful capabilities. Third, our model exhibited task performance and internal representations that were compatible with behavioral and fMRI data from previous human experiments (Kumaran et al., 2012; Park et al., 2021), providing the first theoretical account, to our knowledge, for a cognitive map representation specific to the human hippocampal formation.

Our contributions can be summarized as follows. In neuroscience, we provided a new theory that the principle of abstract relational structure can account for the neural representation in the human hippocampal formation during relational decision-making tasks. In AI research, we identified a novel learning problem that requires a drastic generalization over relations from a small dataset and thus challenges existing powerful models, but can be solved by our proposed model at the human level.

## 2 BACKGROUND: RELATIONAL DECISION-MAKING TASKS

To motivate our specific model design, we briefly review the tasks used in previous human experimental studies on relational learning. In Kumaran et al. (2012), the task is to learn a "one-dimensional hierarchy" or sequential ordering among 7 images of human faces or objects (Figure 1a). The true relations are not made known to the subject. During the learning session, in each trial, the subject is presented with a pair of randomly chosen images that are adjacent in the ordering and required to make decision on the relation between them, either higher or lower; the subject gets rewarded

if the answer is correct (Figure 1c). After a certain number of trials, the subject's acquisition of the relation is assessed in a transitive inference task. This requires the subject to iteratively use the learned relation to relate a given pair of non-adjacent images. The study reported nearly perfect score for this task by the human subjects (Kumaran et al., 2012).

In Park et al. (2021), the task is to learn a "two-dimensional hierarchy" among 16 images of human faces (Figure 1b). The images are assumed to be organized in a $4 \times 4$ grid, which gives two types of sequential ordering among the individuals corresponding to the two axes ("competence and popularity"). The true relations are again unknown to the subject. The task proceeds similarly to the one-dimensional case, except that, in each trial, the subject is hinted with one of the two axes in which to infer the relation between a given pair of images (multi-axis). Also, unlike the one-dimensional case, each image can be related with up to four other images (many-to-many relation). After the learning session, the study conducted an fMRI experiment, which revealed distance coding and hexagonal modulation properties in the hippocampus and entorhinal cortex (Park et al., 2021).

## 3 Framework

### 3.1 Task formalized

Inspired by the human experiments overviewed in Section 2, we formalize our task design for training and evaluation of our proposed model. First, we assume a *relationship* $\mathcal{A}$ consisting of a finite set of relations. We also assume a *domain* $\mathcal{D} = (\mathcal{E}_\mathcal{D}, \mathcal{R}_\mathcal{D})$ with a finite set $\mathcal{E}_\mathcal{D} \subset \mathbb{R}^E$ of concrete entities or simply *entities* and a set $\mathcal{R}_\mathcal{D}$ consisting of tuples $(x_i, a_i, x_i')$ with $x_i, x_i' \in \mathcal{E}_\mathcal{D}$ and $a_i \in \mathcal{A}$. In the sequel, we generally consider a set of domains in which a common relational structure is imposed on the entities. For example, the "ordering" relationship with two relations $\prec$ (prior to) and $\succ$ (next to), as in Figure 1a, has the following general structure. The two relations are converse to each other: if a domain has $(x_1, \prec, x_2)$, then it also has $(x_2, \succ, x_1)$, and vice versa; in the sequel, we sometimes write $x_1 \prec x_2$ instead of $(x_1, \prec, x_2)$. In addition, every domain has a "minimal" entity $x_\perp$ for which there is no $x'$ such that $(x', \prec, x_\perp) \in \mathcal{R}_\mathcal{D}$; similarly, every domain has a "maximal" entity $x_\top$; we call these *terminal* entities.

Our goal is to learn the general relational structure hidden in a given set of domains by performing the following relational inference task over multiple domains. The entire task undergoes a series of epochs (Figure 1d). Each epoch, given some domain $\mathcal{D}$ (unknown to the model), repeats the following process for $T$ times: at time $t$, (i) receive a (random adjacent) pair of entities $x_t, x_t'$, (ii) infer their relation $a_t$, and (iii) obtain an immediate reward $r_t = 1$ if the inference is correct, i.e., $(x_t, a_t, x_t') \in \mathcal{R}_\mathcal{D}$, or no reward $r_t = 0$ otherwise (Figure 1c). In the training phase, the model learns relational representation by performing the above process while maximizing total rewards for training domains. In the test phase, the model performs a similar process for held-out test domains, which have new entities but with the same relational structure; the total rewards give the performance score. Note that it is the test phase that corresponds to the human tasks described in Section 2; the training phase would correspond to the subject's all experience prior to the experiments. Our framework adopts the same task design for both phases for simplicity. Note also that the task structure is different from random-walking (Whittington et al., 2018; 2020), where we receive an entity $x_t$ and a relation $a_t$ in each step and predict the next entity $x_{t+1}$ such that $(x_t, a_t, x_{t+1}) \in \mathcal{R}_\mathcal{D}$.

### 3.2 Model structure

Our model, ARDMO, has an architecture with two modules, one representing abstract relations and the other representing their correspondence to concrete entities (Figure 1e). Although such modular architecture is somewhat similar to TEM (Whittington et al., 2018; 2020) as well as other memory-based models (Graves et al., 2014; 2016; Webb et al., 2020), our main novelty lies in the learning algorithm described in Section 3.3.

To represent abstract relations, we assume abstract entities or *states* $s \in \mathbb{R}^S$. We define the "relational" probability $p(s'|s, a)$ that state $s'$ is related with given state $s$ by given relation $a \in \mathcal{A}$:

$$p(s'|s, a) = \mathcal{N}(\rho(R_a s), \sigma^2 I) \tag{1}$$

where $R_a \in \mathbb{R}^{S \times S}$ is a *relation matrix* specific to relation $a$ and $\sigma$ is a (global) scalar; $\rho$ is an activation function, for which we use the $L_2$-normalization $\rho(s) = s/\|s\|$. We also define the "prior"

probability $p(a|s)$ that given state $s$ has relation $a$ (with some other state):

$$p(a|s) = g_a(s) \quad \text{where} \sum_a g_a(s) = 1 \tag{2}$$

where $g_a(s)$ is a non-linear function specific to $a$. The prior distribution is particularly important to represent terminal entities, e.g., $p(\prec |s) \approx 0$ for maximal entities.

To represent binding between abstract states and concrete entities, we introduce a memory mechanism. Assuming a *key matrix* $W \in \mathbb{R}^{H \times S}$ and a *memory matrix* $M \in \mathbb{R}^{E \times H}$, we refer to an entity corresponding to a state $s$ by the following function:

$$\text{read}_{M,W}(s) = Mh \quad \text{where} \ h = \text{softmax}(Ws) \tag{3}$$

Here, the intermediate variable $h$ softly represents an address pointing to the content of an entity stored in the memory. As discussed below, we take the key matrix $W$ as a parameter, but the memory matrix $M$ as a hidden variable. The latter allows for dynamical updates of the binding, which is crucial for discovering domain-general relational structure.

## 3.3 LEARNING ALGORITHM

Given a set $\mathcal{D}_1, \ldots, \mathcal{D}_D$ of domains, our learning procedure for ARDMO runs a series of epochs described as follows (Figure 1e). In each epoch, we start with randomly selecting a domain $\mathcal{D}_d$ and randomly initializing the memory matrix $M_1$. At each time $t$, given a pair of entities $x_t, x_t'$, we first obtain the corresponding states $s_t = \text{infer}_{M_t,W}(x_t)$ and $s_t' = \text{infer}_{M_t,W}(x_t')$ using the following "content-based" inference function:

$$\text{infer}_{M,W}(x) = W^\mathsf{T} h \quad \text{where} \ h = \text{softmax}(M^\mathsf{T} x) \tag{4}$$

After this, we make decision on the relation between these by sampling:

$$\tilde{a}_t \sim p(a_t|s_t, s_t') \tag{5}$$

where the "posterior" distribution $p(a_t|s_t, s_t')$ can be obtained by Bayes' rule using equations 1 and 2. The reward given for the decision is $r_t = 1$ if $(x_t, \tilde{a}_t, x_t') \in \mathcal{R}_{\mathcal{D}_d}$, or $r_t = 0$ otherwise. Note that we use the memory to estimate both states ($s_t$ and $s_t'$) and use the relational representation to infer their relation ($a_t$), unlike TEM (Whittington et al., 2018; 2020) or other ordinary recurrent networks, which use a relational representation to estimate the next state ($s_t$) from given previous state ($s_{t-1}$) and relation ($a_t$).

In the rewarded case, we adjust the memory so as to integrate the given input entities and the current relational representations and thereby reflect the inference result. For this, we first re-postulate that the second state comes from the distribution for the inferred relation:

$$\tilde{s_t'} \sim p(s_t'|s_t, \tilde{a}_t) \tag{6}$$

(but leave the first state $s_t$ as it is for simplicity). We then recall the current memory contents $\tilde{x}_t = \text{read}_{M_t,W}(s_t)$ and $\tilde{x_t'} = \text{read}_{M_t,W}(\tilde{s_t'})$ and simultaneously update the memory:

$$M_{t+1} \leftarrow M_t + \alpha \left[ (x_t - \tilde{x}_t)\text{softmax}(Ws_t)^\mathsf{T} + (x_t' - \tilde{x_t'})\text{softmax}(W\tilde{s_t'})^\mathsf{T} \right] \tag{7}$$

In the update, we use a relatively large coefficient, e.g., $\alpha = 0.7$, which results in a drastic change of the memory contents from the old ones ($\tilde{x}_t$ and $\tilde{x_t'}$) to the new ones ($x_t$ and $x_t'$). Note that our memory mechanism is rather different from Hopfield-type auto-associative memory like Ba et al. (2016), which is used in TEM (Whittington et al., 2018; 2020), where the update rule only stores new associations, not erasing old ones. In this sense, our memory is more similar to Graves et al. (2014; 2016), although the old content is estimated directly from the memory in our update rule (equation 7), whereas it is computed by the controller network in their case. As shown in the experiment in Section 4.1, the choice of memory mechanism is essential in our setting. This is probably because our task induces crucial interaction between states and memory and thus earlier inaccurate state-entity associations must be replaced later. (No such problem seems to arise in TEM since current states can often be inferred directly from previous states in the random-walk task.)

Then, we define the following one-step loss function, which encourages correct inferences and discourage incorrect ones:

$$\mathcal{L}_t = (r_t - p(\tilde{a}_t | s_t, s'_t))^2 \tag{8}$$

Our goal is to minimize this over steps and epochs with respect to the parameters $\Phi = \{R_a, \phi(g_a) | a \in A\} \cup \{W, \sigma\}$, where $\phi(\cdot)$ is the set of parameters used in the given function. Note that the loss function depends on the states, which in turn depend on the memory, which further depends on the previous memory, inputs, etc. Therefore optimizing the loss necessarily causes back-propagation through time.[2] Note that recurrent computation here occurs primarily through memory rather than through states, which is somewhat unique to our model. (Our algorithm can thus be seen to optimize the procedure of how to update the memory, thus "learning to learn" the relations.)

Lastly, we incorporate a simple regularization that enforces the prior to follow the actual occurrences of the relations.

$$\mathcal{L}_t^{\texttt{prior}} = (r_t - p(\tilde{a}_t | s_t))^2 \tag{9}$$

In our experience, without this regularization, the learned prior tends to be uniform. This makes the model fail to capture terminal entities and thereby degrade performance in transitive inference (Section 4.1). The entire learning procedure is summarized as pseudo-code in Figure 5.

### 3.4 TRANSITIVE INFERENCE

Transitive inference is an excellent task to test the learned relational representation. In this, given a pair $x, x'$ of entities, we ask to infer the relation $a$ whose one or more iterate, or *transitive closure*, relates them: $(x, a, x_1), (x_1, a, x_2), \ldots, (x_{m-1}, a, x) \in \mathcal{D}$ for some $m \geq 1$ (where the subscript is for iteration, not for time).

To solve this, we first obtain the states $s, s'$ corresponding to $x, x'$. We then calculate the following two probabilities $p^+(s'|s, a, m)$ (that state $s'$ is related with given state $s$ by the $m$-th iterate of given relation $a$) and $p^+(a|s, m)$ (that given state $s$ has $m$-th iterate of relation $a$):

$$p^+(s'|s, a, m) = p(s'|\psi_a^{m-1}(s), a) \qquad p^+(a|s, m) = \prod_{i=0}^{m-1} p(a|\psi_a^i(s)) \tag{10}$$

where $\psi_a(s) = \kappa(\rho(R_a s))$ and $\kappa(s) = \text{infer}_{M,W}(\text{read}_{M,W}(s))$; $\psi^m$ means the $m$-th iteration of function $\psi$. Using these, we next obtain the following probability $p^+(a|s, s')$ (that given states $s$ and $s'$ are related by the transitive closure of relation $a$):

$$p^+(a|s, s') = \frac{1}{M} \sum_{m=1}^{M} \frac{p^+(s'|s, a, m) p^+(a|s, m)}{\sum_{a'} p^+(s'|s, a', m) p^+(a'|s, m)} \tag{11}$$

(assuming that only the same relation is iterated at most $M$ times). Finally, we answer $a^+ = \arg\max_a p^+(a|s, s')$ for the relation in question. For later evaluation, we also use the confidence value $c^+ = \max_a p^+(a|s, s')$.

The above approach can actually be derived as an approximation in a certain probabilistic framework (Appendix A). The approach works in the case of one-to-one relations as used in our experiment in Section 4.1. Note that, for successful transitive inference, it is crucial for the model to learn both distributions $p(s'|s, a)$ and $p(a|s)$ precisely; otherwise, it may make a wrong judgment, e.g., that a state has the $m$-th iterate of $a$ even if it does not.

## 4 EXPERIMENTS

In this section, we present the results of experiments on two tasks. Additional details on the experiments are given in Appendices B and C.

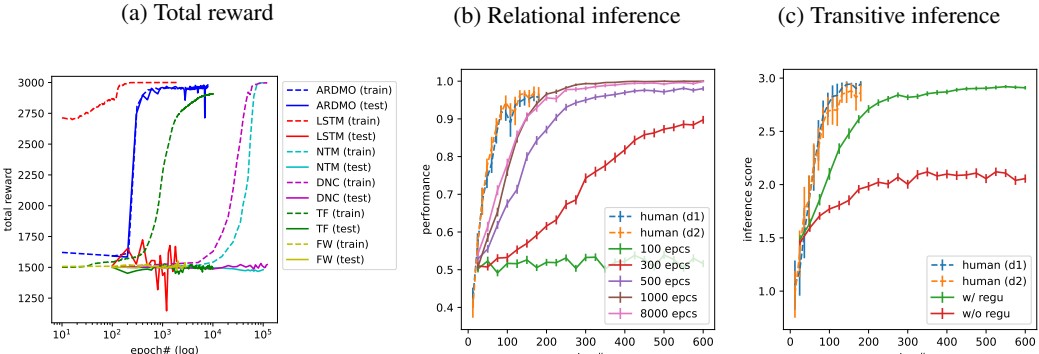

Figure 2: Results on the one-dimensional hierarchy task. (a) Traces of total rewards for ARDMO for training and test data, contrasted with LSTM, NTM, DNC, Transformer (TF), and Fast-Weights-based ARDMO (FW); FW can be seen as a version of TEM. The horizontal axis is log-scaled. (NTM and DNC took very long training time $\sim 40$ days). (b) Performance ramp in the relational inference within an epoch (up to 600 steps) at different training stages, together with the corresponding human data (for two domains). (c) Inference score in the transitive inference for the models trained with and without regularization, together with human data. The human data are replotted from Kumaran et al. (2012). Error bars show standard errors over domains (models) or subjects (humans).

## 4.1 ONE-DIMENSIONAL HIERARCHY

The first experiment here simulates the previous human experiment on a simple one-dimensional hierarchy (Kumaran et al., 2012) outlined in Section 2. Using CIFAR100, we formed each domain consisting of 7 images as entities, randomly selected and ordered from a single, randomly chosen object class. We prepared 600 training domains and 100 test domains so that these included images from disjoint classes. We assumed two (ordering) relations $\prec$ and $\succ$ as introduced in Section 3.1. Recall that these relations are defined only for adjacent pairs; there are minimal and maximal entities. Note also the small-size nature of the task: only 12 possible pairs per domain. In the training phase, starting with a randomly initialized instance of ARDMO ($H = 50$; $S = 20$), we ran 8000 epochs each performing 3000 training steps for a randomly chosen domain. The training took about 3 days. We then proceeded to the test phase, which was similar to training, but without optimization with respect to the model parameters. Below, we present the results.

First, Figure 2a shows the trace of per-epoch total rewards during training and test for a model instance. The training succeeded with almost full rewards. Importantly, the test followed a similar trace to the training, indicating that the model successfully generalized the relational structure to unseen entities. Figure 2a also gives comparison with other existing models trained on the same task (see Appendix B.2.2 for details). (1) LSTM (Hochreiter and Schmidhuber, 1997), a conventional recurrent model, showed no such generalization, which is no surprise since it has no updatable memory. (2) NTM (Graves et al., 2014) and DNC (Graves et al., 2016), recurrent models with memory mechanism, also showed no generalization, despite their complex and powerful capability. This can be because that these models do not decouple well abstract and concrete representations (e.g., values to write in to memory are determined only by the recurrent network). (3) vanilla Transformer (Vaswani et al., 2017) similarly failed to generalize; we tried various architecture and training settings but got similar results. We consider that the model structure is too complex to match our task nature with very small dataset and high-dimensional input.[3] (4) A modified version of ARDMO using Fast Weights, a Hopfield-type memory mechanism (Ba et al., 2016), failed even in training. The model can be seen as a version of TEM (Whittington et al., 2018; 2020) replacing the learning algorithm

---

[2]Note that the learning procedure involves discrete sampling (equation 5), which prevents some gradients from being propagated. Empirically, however, our algorithm stably optimizes (Section 4), possibly using imprecisely computed gradients. We observed no notable improvement when using a straight-through estimator (Bengio et al., 2013; Jang et al., 2017).

[3]Whittington et al. (2022) shows that Transformer can accommodate a version of TEM, for which no such optimization problem happens probably because they restrict weight matrices in a particular way.

with ours (to match the task format), but retaining the memory mechanism. The shown result thus implies that the appropriate choice of memory mechanism highly depends on the task structure. In sum, our comparisons above highlight the particular difficulty of our task that completely defeats these powerful models.

Second, we inspected how our model behaves within an epoch on test data. Figure 2b shows the trace of per-block test performance (probability of reward in a block of 25 steps) in the relational inference within an epoch at different stages of training. In each epoch, the model started to perform badly but gradually became better. This behavior is expected since the model initially knew nothing about new entities but got acquainted with their association with the learned relation during the epoch. Indeed, as the representation got improved during training, the time to gain full rewards (thus identify the correct relations) became shorter and shorter. After completion of training (8000 epochs), the peak performance was comparable with the human data (Kumaran et al., 2012) replotted in Figure 2b. (We are not concerned here about the speed of ramp as human brains can clearly do much more complex operations in a single trial, such as replays and some kind of inferences—the model can at best simulate human at the abstract level.)

Third, we tested how well our model could exploit the learned representation for transitive inference task, using the scheme described in Section 3.4. For the sake of comparison with human data, we calculated the inference score, namely, the performance multiplied by the confidence value ($\lfloor 2.99c^+ + 1 \rfloor$). Figure 2c shows that the inference score ramped in accordance with the relational inference in Figure 2b. Again, the peak score was comparable to the corresponding human data (Kumaran et al., 2012), replotted in Figure 2c. In addition, Figure 2c shows that, when a model was trained without the regularization described in Section 3.3 and therefore did not precisely learn the prior distribution, the performance became significantly poorer. This result suggests that, in the human experiment, although the task did not explicitly require to infer the terminalities of entities, humans might have recognized these implicitly and thereby achieved the high task score. This result can stand as a testable prediction.

Finally, we confirmed that the results were robust across model instances trained under the same condition or under some hyperparameter variations, though the choice of a high update rate ($\alpha$) was crucial. Appendix B.3 summarizes the results.

## 4.2 TWO-DIMENSIONAL HIERARCHY

The second experiment simulates the previous human fMRI experiment on the two-dimensional hierarchy task (Park et al., 2021), as outlined in Section 2. The main aim of the human experiment was to investigate the neural representation of a non-spatial 2D map in the human hippocampal formation. In particular, they reasoned that, if the 2D map is represented by a neural population with hexagonal grid fields (Hafting et al., 2005), the aggregate neural responses measured by fMRI should overall increase in particular directions in the map and decrease in other directions and there should be equally spaced six directions with increase (Figure 4); this property is called hexagonal modulation.

Our interest is whether hexagonal modulation emerges in our proposed model. Thus, we formed training and test domains each consisting of 16 images as entities, analogously to the 1D case. As noted in Section 2, the task involved multi-axis, many-to-many relations. That is, the 16 entities intendedly formed $4 \times 4$ grid, with ordering in two axes: $\prec_1$ and $\succ_1$ in axis 1, and $\prec_2$ and $\succ_2$ in axis 2 (Figure 1b). Also, in one axis, each entity can be related with up to four entities; e.g., $x \prec_1 x'$ can hold for entities that are adjacent in axis 1, but potentially non-adjacent in axis 2. We trained an instance of ARDMO ($H = 500, S = 20$) for 6000 epochs, with 2000 steps per epoch. In each epoch, a target axis $i$ was randomly chosen and given to the model so that it could infer between $\prec_i$ and $\succ_i$. The training conditions were otherwise similar to Section 4.1. For comparison, we trained another model instance with smaller memory ($H = 50$) and a null model (trained with the same condition but random rewarding).[4] The training of each model took about 2 days. As a basic assessment, we ran the two trained models in the relational inference task in test domains and confirmed their performance ramps to a reasonably high score for both models (Figure 8ab).

---

[4]Since no other model so far has succeeded in solving our task and neural plausibility of a non-working model would be meaningless, we provide here no answer as to whether or not the similarity with the human fMRI data is unique to ARDMO.

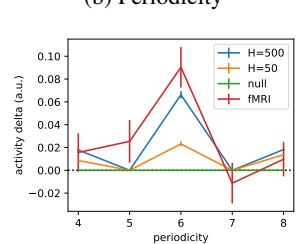
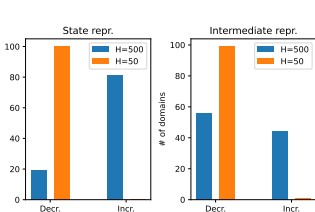

Figure 3: Results on the two-dimensional hierarchy task. (a) The hexagonal modulation property. (b) The estimated magnitude at each periodicity. (c) Distance coding in the state (left) or intermediate (right) representations. The shown results are for ARDMOs with larger ($H = 500$) or smaller memory ($H = 50$) in addition to a null model and human fMRI data replotted from Park et al. (2021) (Fig. 4b). Error bars show standard errors over domains (models) or participants (humans).

To see if the learned model showed hexagonal modulation, we performed the following analyses after running the model in each test domain (Appendix C.1 for details). To simulate the protocol in Park et al. (2021), where they measured the average brain activities during consecutive presentation of image stimuli at each pair of grid positions, we obtained the state vectors ($s$) corresponding to the images (using equation 4) and took their mean. The (mean) state values, when plotted against the grid direction (angle) between the two positions, overall showed a periodic pattern of increase and decrease in the cycle of $60\deg$; Figure 3a plots the state values that were phase-aligned (across dimensions) and averaged (over dimensions and domains, within a bin of $30\deg$ width), which indicates hexagonal modulation. To quantify the magnitude of the modulation, we calculated the mean differences between the maximal and the minimal of average state values (6-fold periodicity) and compared it with the cases of 4-, 5-, 7-, and 8-fold periodicities; the 6-fold periodicity gave the largest magnitude (Figure 3b). The results resemble the human fMRI data (Park et al., 2021) (replotted in Figure 3ab). The model with smaller memory also gave similar results but with a lower magnitude.

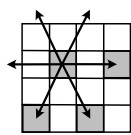

Figure 4: Modulatory map directions

In addition, Park et al. (2021) observed particular distance coding of a 2D map in the hippocampus and entorhinal cortex, where representations tended to be more similar between closer grid positions. To conduct a compatible analysis on our learned model, we obtained state and intermediate vectors ($s$ and $h$, using equation 4) corresponding to the images at each pair of grid positions and computed the dissimilarity (Euclidean distance) between the vectors (Appendix C.1). Repeating these for different domains, the dissimilarity had either increasing or decreasing tendencies against the (Euclidean) distance in the grid space. Figure 3c summarizes the number of domains with increase or decrease in the intermediate or state representation. In both representations, increasing tendencies were rarely found in the model with smaller memory, but much more often with larger memory, thus more similar to the human fMRI data reporting only increasing tendencies (Park et al., 2021). These results are robust across model instances trained in the same condition (Appendix C.2).

As mentioned, hexagonal modulation is expected as mixed neural signals from cells in the human entorhinal cortex if they have hexagonal grid fields as in the rodent (Hafting et al., 2005). In this sense, our result reproducing hexagonal modulation is related to previous model studies reproducing the grid cell property such as Whittington et al. (2018; 2020). However, note that neither result is directly predictable from the other since the task and model structures are rather different. Also, our study only offers a high-level (abstract) account of the entorhinal cortex representation and therefore how it connects to the cell-level grid-cell property remains open.

## 5 RELATED WORK

Recently, learning models for the cognitive map representations in the hippocampal formation have been drawing attention. Among others, Whittington et al. (2018; 2020) proposed TEM, which accounts for place and grid cells in rodents as an abstract relational structure of 2D-geometric environments while performing spatial random-walking tasks. Our study has been much influenced

by theirs, adopting a similar modular architecture with separate relation and memory representations. However, the different task goal, relational decision-making, necessitates our model construction to use different approaches in (1) the learning algorithm that maximizes rewards from memory-based inference and (2) the memory mechanism with the update rule that can replace old contents. In particular, we showed that the Hopfield-type memory proposed by Ba et al. (2016) does not work in our setting (Section 4.1), which is quite striking, given that TEM was successful by using this mechanism. Thus, a simplistic adaption of TEM to our task fails. Later, Whittington et al. (2022) simplified TEM and clarified a formal relationship with Transformers (Vaswani et al., 2017). A further refinement of these models revealed minimal constraints to reproduce grid-cell properties (Dorrell et al., 2022). A similar modular architecture for learning general 2D structure has also been proposed by Uria et al. (2022), explaining a variety of hippocampal neural properties.

Different approaches can also explain cognitive-map properties from sequential learning. Clone-structured cognitive graphs allow for learning graph representations that are abstracted from concrete observations (George et al., 2021; Raju et al., 2022; Guntupalli et al., 2023), somewhat similarly to TEM. The learned representations, in conjunction with planning through probabilistic inference, reproduce various complex hippocampal properties like remapping. However, the model offers no account for entorhinal cortex properties due to the finite, discrete nature of state space. Other accounts of the hippocampal formation have also been proposed using intermediate representations of recurrent networks (Banino et al., 2018; Sorscher et al., 2019), successor representations of reinforcement learning (Stachenfeld et al., 2017), or simple unsupervised learning (Dordek et al., 2016).

Memory-augmented recurrent models have been used for solving general, complex tasks requiring relational reasoning. Some such models use memory for connecting abstract relations with concrete entities to discover abstract rules in input sequences and thereby solve symbol-processing tasks (Webb et al., 2020; Chen et al., 2021). However, despite the apparent similarity, their tasks to find out common rules hidden within fixed-length sequences are not compatible with our tasks to find out general structure in a set of binary relations. Graves et al. (2014; 2016) have proposed recurrent networks with external memory, highly influenced by von Neumann machines, that can learn to solve list and graph problems. Although these models are powerful, their way of coupling the recurrent network and the memory module seems to deteriorate disentangled representation of abstract relations and concrete domains (Section 4.1). Santoro et al. (2018) has presented a different approach for relational reasoning by using multi-head attention that allows for interaction between memory slots.

Recent studies have incorporated memory mechanisms in classifier neural network (Ba et al., 2016), in reinforcement learning (Pritzel et al., 2017; Hansen et al., 2018; Fortunato et al., 2019), in generative models (Li et al., 2016; Bornschein et al., 2017; Wu et al., 2018), in meta-learning (Santoro et al., 2016; Munkhdalai and Yu, 2017), and so on. Although these models have some technical commonalities with ours, they use memory to efficiently recall past experience and thereby increase specific task performance; therefore the goals are largely different from ours.

## 6 CONCLUSION

In this study, we have proposed a novel memory-based cognitive model for learning abstract relational structure from decision-making tasks. The results showing good match with previous human behavioral and fMRI data indicate that the principle of abstract relational structure might account for the hippocampal representations not only for spatial random-walking but also for more general relational tasks. In another view, the present study addressed the challenging AI problem of cross-domain generalization from a few examples, highlighting the crucial role of memory in abstract conception. Various future directions are conceivable: in AI research, representations of general graph-like relational structures and learning with delayed or implicit rewards; in neuroscience, accounts for other hippocampal phenomena like replays, interaction with other brain areas like the prefrontal cortex, and connection with rodent data.

ACKNOWLEDGMENTS

This work has been supported by Grants-in-Aid for Scientific Research (21K19812), Okawa foundation (22-06), New Energy and Industrial Technology Development Organization (P20006), and Corporate Sponsored Research Programs for World Model and Simulator of The University of Tokyo.

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

## A  DERIVATION OF THE APPROXIMATION IN SECTION 3.4

For solving the transitive inference task, we consider the probability $p^+(s'|s, a, m)$ that state $s'$ is related with given state $s$ by the $m$-th iterate of given relation $a$:

$$p^+(s'|s, a, m) = \iint \cdots \int \left[ \prod_{i=0}^{m-2} p(s_{i+1}|s_i, a) \right] p(s'|s_{m-1}, a) ds_1 ds_2 \ldots ds_{m-1} \quad (12)$$

and the probability $p^+(a|s, m)$ that given state $s$ has $m$-th iterate of relation $a$:

$$p^+(a|s, m) = \iint \cdots \int \left[ \prod_{i=0}^{m-2} p(s_{i+1}, a|s_i) \right] p(a|s_{m-1}) ds_1 ds_2 \ldots ds_{m-1} \quad (13)$$

The above definitions are intractable in general. However, assuming one-to-one relations, we use the following approximation:

$$p^+(s'|s, a, m) \approx p(s'|\psi_a^{m-1}(s), a) \qquad p^+(a|s, m) \approx \prod_{i=0}^{m-1} p(a|\psi_a^i(s)) \quad (14)$$

where $\psi_a(s) = \rho(R_a s)$. (Empirically, we find it more useful to "clean up" the state after each iteration, and therefore use $\psi_a(s) = \kappa(\rho(R_a s))$ with the function $\kappa(s) = \text{infer}_{M,W}(\text{read}_{M,W}(s))$, which recalls the concrete entity of a given state vector and then mapping it back to a state.)

To derive the above, we crudely approximate the Gaussian distribution in equation 1 by a delta function: $p(s'|s,a) \approx \delta[s' = \rho(R_a s)]$. To derive the approximation of $p^+(s'|s,a,m)$ in equation 14, first let:

$$A_{m,a}^k = \iint \cdots \int \left[ \prod_{i=0}^{m-2-k} p(s_{i+1}|s_i,a) \right] p(s'|\psi_a^k(s_{m-1-k}),a) ds_1 ds_2 \ldots ds_{m-1-k} \quad (15)$$

For $0 \leq k \leq m-2$, by the approximation assumption $p(s'|s,a) \approx \delta[s' = \psi_a(s)]$, we obtain:

$$A_{m,a}^k \approx \iint \cdots \int \left[ \prod_{i=0}^{m-3-k} p(s_{i+1}|s_i,a) \right] \delta[s_{m-1-k} = \psi_a(s_{m-2-k})]$$
$$p(s'|\psi_a^k(s_{m-1-k}),a) ds_1 ds_2 \ldots ds_{m-1-k} \quad (16)$$

$$= \iint \cdots \int \left[ \prod_{i=0}^{m-3-k} p(s_{i+1}|s_i,a) \right] p(s'|\psi_a^{k+1}(s_{m-2-k}),a) ds_1 ds_2 \ldots ds_{m-2-k} \quad (17)$$

$$= A_{m,a}^{k+1} \quad (18)$$

Thus, noting $A_{m,a}^0 = p^+(s'|s,a,m)$, the result follows.

To derive the approximation of $p^+(a|s,m)$ in equation 14, let:

$$B_{m,a}^k = \iint \cdots \int \left[ \prod_{i=0}^{m-2-k} p(s_{i+1},a|s_i) \right] \prod_{i=0}^{k} p(a|\psi_a^i(s_{m-1-k})) ds_1 ds_2 \ldots ds_{m-1-k} \quad (19)$$

For $0 \leq k \leq m-2$, by the same approximation assumption, we obtain:

$$B_{m,a}^k \approx \iint \cdots \int \left[ \prod_{i=0}^{m-3-k} p(s_{i+1},a|s_i) \right] \delta[s_{m-1-k} = \psi_a(s_{m-2-k})] p(a|s_{m-2-k})$$
$$\left[ \prod_{i=0}^{k} p(a|\psi_a^i(s_{m-1-k})) \right] ds_1 ds_2 \ldots ds_{m-1-k}$$
$$(20)$$

$$= \iint \cdots \int \left[ \prod_{i=0}^{m-3-k} p(s_{i+1},a|s_i) \right] \left[ \prod_{i=0}^{k+1} p(a|\psi_a^i(s_{m-2-k})) \right] ds_1 ds_2 \ldots ds_{m-2-k} \quad (21)$$

$$= B_{m,a}^{k+1} \quad (22)$$

Noting $B_{m,a}^0 = p^+(a|s,m)$, the result follows.

## B  ADDENDUM ON THE EXPERIMENT IN SECTION 4.1

### B.1  DATASETS

In each experiment, we prepared a dataset using the CIFAR100 image database. To form each domain, we randomly chose an object class from which we randomly selected 7 images as entities and then ordered them randomly. We formed 600 training domains and 100 test domains, with no overlapping object classes between the two. Each image was compressed to 700 dimensions by PCA and normalized to unit norm.

### B.2  TRAINING

#### B.2.1  ARDMO

For the architecture, apart from the description in Section 3, we used a two-layer perceptron to implement $g_a(s)$ (for prior) where the intermediate layer had 10 units with the sigmoid activation function and the output layer was fed to the softmax function. The pseudo-code of the learning algorithm is given in Figure 5.

---

1: **procedure** LEARN($\mathcal{D}_1, \mathcal{D}_2, \ldots, \mathcal{D}_D, N$)
2:     initialize $\Phi = \{R_a, \sigma, \phi(g_a) | a \in A\} \cup \{W\}$
3:     **for all** $n = 1, \ldots, N$ **do**
4:         randomly select $d \in [1, D]$
5:         initialize $M_1$
6:         **for all** $t = 1, \ldots, T$ **do**
7:             observe $(x_t, x_t')$ in $\mathcal{R}_{\mathcal{D}_d}$
8:             $s_t = \text{infer}_{M_t, W}(x_t)$
9:             $s_t' = \text{infer}_{M_t, W}(x_t')$
10:            sample $\tilde{a}_t \sim p(a_t | s_t, s_t')$
11:            $r_t \leftarrow \text{get\_reward}((x_t, \tilde{a}_t, x_t'), \mathcal{R}_{\mathcal{D}_d})$
12:            **if** $r_t = 1$ **then**
13:                sample $\tilde{s_t'} \sim p(s_t' | s_t, \tilde{a}_t)$
14:                $\tilde{x}_t = \text{read}_{M_t, W}(s_t)$
15:                $\tilde{x_t'} = \text{read}_{M_t, W}(\tilde{s_t'})$
16:                $M_{t+1} \leftarrow M_t + \alpha(x_t - \tilde{x}_t)\text{softmax}(Ws_t)^\mathsf{T} + \alpha(x_t' - \tilde{x_t'})\text{softmax}(W\tilde{s_t'})^\mathsf{T}$
17:     $\mathcal{L} = \sum_{t=1}^{T} (r_t - p(\tilde{a}_t | s_t, s_t'))^2 + \gamma (r_t - p(\tilde{a}_t | s_t))^2$
18:     miminize $\mathcal{L}$ w.r.t. $\Phi$

---

Figure 5: Learning algorithm for ARDMO

To train the model, we used hyper-parameters $\alpha = 0.7$ and $\gamma = 1$, and Adam optimizer (Kingma and Ba, 2015) with mini-batch size 5. We also used the same $\alpha = 0.7$ for test. To boost the learning, we used a truncated back-propagation strategy. More precisely, we ran optimization after every 25 steps in an epoch. That is, after every 25 steps, we first initialized the accumulated loss function and stopped gradient propagation for the memory matrix $M$.

Generally, from observation of the loss function, training tended to proceed as follows. Initially, learning started with a lengthy plateau (where the loss scarcely changed), then suddenly shifted to a rapid drop (where the loss quickly improved), and thereafter fell into another plateau. This stairway-like course continued for a few times and finally reached a very long and slow slope for convergence. In some cases, the initial plateau was so long that it was unclear whether the optimization simply failed, in which case we manually stopped it after around 2000 epochs and started it over.

### B.2.2 BASELINE MODELS

We adopted the following baseline models.

**Long short-term memory (LSTM) (Hochreiter and Schmidhuber, 1997)** A well-known recurrent model. Input dimensions: 700. Number of layers: 1. Hidden state dimensions: 70. Batch size: 5.

**Neural Turing Machine (NTM) (Graves et al., 2014)** A recurrent model with updatable external memory. Input dimensions: 700. Hidden state dimension: 20. Memory height (number of slots): 50. Memory width (content dimensions): 100. Batch size: 5.

**Differentiable Neural Computer (DNC) (Graves et al., 2016)** An extension of NTM. Input dimensions: 700. Hidden state dimension: 20. Memory height: 50. Memory width: 100. Number of write heads: 1. Number of read heads: 4. Batch size: 5.

**vanilla Transformer (Vaswani et al., 2017)** A well-known language model incorporating self-attention with (additive) sine/cosine position-encoding and temporal masking. Input dimensions: 100. Number of layers: 2 (repeated). Number of heads: 5. Hidden state dimension: 2048. Drop-out rate: 0.1. Context size: 100. Batch size: 100.

**Fast-Weight-based ARDMO** A modified version of ARDMO using the auto-associative memory proposed in Ba et al. (2016). The encoding/decoding method of state and content described

Table 1: Summary of results of relational inference (performance) and transitive inference (inference score) on one-dimensional hierarchy (at 1000 steps in each epoch). The mean and s.d. of scores over 8 model instances are shown.

|  | ARDMO | ARDMO w/o reg. | LSTM | TF | FW | Human (d1) | Human (d2) |
|---|---|---|---|---|---|---|---|
| Relational Inf. | $1.00 \pm 0.00$ | $1.00 \pm 0.00$ | $0.51 \pm 0.02$ | $0.50 \pm 0.01$ | $0.51 \pm 0.01$ | 0.96 | 0.97 |
| Transitive Inf. | $2.86 \pm 0.07$ | $2.09 \pm 0.14$ | — | — | — | 2.95 | 2.87 |

in Whittington et al. (2020, Section 4) was used. Input dimensions: 100. (Other architecture parameters were the same as ARDMO.)

We used the default Pytorch implementations of LSTM and Transformer and a third party implementation of NTM and DNC[5].

In all of these, we input the concatenation of two given images to the model at each step and the correct relation as target. For NTM and DNC, we appended the correct relation in one step to the input in the next step, so that the model could take the correctness of the previous inference into account. For Transformer, we gave the concatenated image pair and the target alternately in the input and let the model infer the target from each image pair, whose summed squired error became the loss function. For Fast-Weight-based model, we chose the reduced input dimensions (100) since the required memory size, quadratic in the input dimensions, would become prohibitive for the original dimensions (700); we chose the fixed update rate $\eta = 0.5$ following Ba et al. (2016).

### B.3 ADDITIONAL RESULTS

Table 1 summarizes the results from 8 model instances for each task and for each model (except NTM and DNC for which we could train only one instance due to the very long training time), together with the human data. Figure 6ab shows the traces of performance in relational inference and inference score in transitive inference for all models in the same format as Figure 2bc (showing model #1).

We also varied two hyperparameters, namely the memory update rate $\alpha$ and the state dimension $S$, and measured the total rewards (for test data). For this, we additionally trained 4 model instances for $\alpha = 0.01, 0.1, 0.3, 0.5, 0.9$ and for $S = 5, 10, 50, 100$. Figure 6c plots the effect of varying $\alpha$, indicating that the performance stayed high for a relatively large value of $\alpha$ (e.g., $\geq 0.5$) but dropped significantly for a smaller value. Figure 6d plots the effect of varying $S$, showing stable performance over the tested range. However, using a further larger state dimension like $S = 200$ often led to failure of training.

For Transformer, we tried a number of other architecture parameter settings than described in Appendix B.2.2, using more layers (5), less heads (1 or 2), or larger context sizes (300). The results were similar: all models failed to generalize. This result was somewhat surprising since the task initially appeared to be easy for Transformer. Indeed, one can imagine a Transformer that uses the input directly as a query and key and then uses position encoding to find the corresponding relation found in the next entry in the sequence. However, optimization never found out such solution but the weights seemed to excessively adapt to the input images, thus leading to overfitting. We also observed that even training did not succeed for higher input dimensions (700) or smaller batch size (20).

Lastly, we examined the ideal observer, which performs as well as possible in a given epoch, assuming that it knows perfectly the 1D task structure. We implemented the ideal observer such that it memorizes all past pairs in the epoch and, at each step, simply searches for the given pair or its flipped pair; if it finds, then it can give the right answer (reward 1) or otherwise a random answer (reward 0.5). As a result, we found that both the model and human were much slower than the ideal observer, which reached nearly 1.0 around 50 steps in our calculation.

---

[5]https://github.com/jingweiz/pytorch-dnc

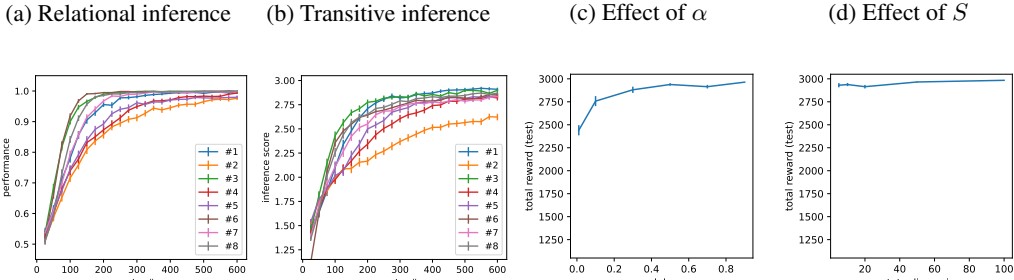

Figure 6: Additional results on one-dimensional hierarchy. (a,b) The traces of performance in relational inference and inference score in transitive inference for 8 ARDMO instances. The error bars are standard errors over the domains. (c,d) The total rewards (test data) by ARDMO instances trained with different update rates or different state dimensions. The error bars are standard errors over the model instances.

## C  ADDENDUM ON THE EXPERIMENT IN SECTION 4.2

### C.1  ANALYSIS METHOD

Our aim here is to analyze the internal representations in our trained model in a way comparable to the brain analysis described in Park et al. (2021).

First, in the fMRI experiment (Park et al., 2021), they analyzed the hexagonal modulation property of the brain signals from the entorhinal cortex during two consecutive visual stimuli corresponding to two grid positions. To simulate this, after running the model in a given test domain, we obtained the state vector corresponding to the image at each grid position by using inference through the memory (equation 4) and took the average over the two state vectors. The values in state values are normalized as z-scores. By plotting the state values in each dimension against the grid direction, i.e., the angle between the grid positions (Figure 7), we obtained a function over angles. Repeating this for different dimensions, we obtained a set of such functions. To see if those functions had periodicity in a given cycle ($90 \deg$, $72 \deg$, $60 \deg$, $51.4 \deg$, and $45 \deg$ corresponding to 4-, 5-, 6-, 7-, and 8-fold periodicities), we visualize the average over the functions, where we needed to deal with that each function may have a different phase. In Park et al. (2021), to avoid contamination of phase and periodicity, they estimated the phases by using another dataset obtained from a separate experiment with the same task condition. We simulated this by using the state vectors obtained through the model at 25 steps prior to the completion of running in the test domain (which has a different memory matrix). To estimate the phase, following the method in Park et al. (2021), we fit state values in each dimension against the sin and cos of the angle and took the arc-tangent of the regression coefficients. We then calculated the circular mean over the obtained phases. Finally, we took the phase-aligned average of the functions for all dimensions separately for each domain, and then averaged the resulting functions for all domains.

Second, in the same experiment Park et al. (2021), they analyzed the distance coding in the brain signals from the hippocampus and entorhinal cortex. To simulate this, we similarly obtained the intermediate or state vectors corresponding to the images at each pair of grid positions (equation 4) and calculated the distance between these as their dissimilarity. Although (Park et al., 2021) used Mahalanobis distance, we used Euclidean distance since our case uses the deterministic state inference. By plotting the dissimilarity against the grid distance, i.e., Euclidean distance between the grid positions (Figure 7), we determined the increase or decrease tendency by the correlation coefficient (positive or negative).

### C.2  ADDITIONAL RESULTS

We trained 8 ARDMO instances with larger ($H = 500$) or smaller ($H = 50$) memory. Figure 8 summarizes the results for these in the same format as Figure 3 (showing model #1). These show overall robustness of the results across model instances.

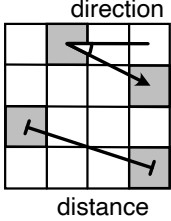

Figure 7: Grid coding

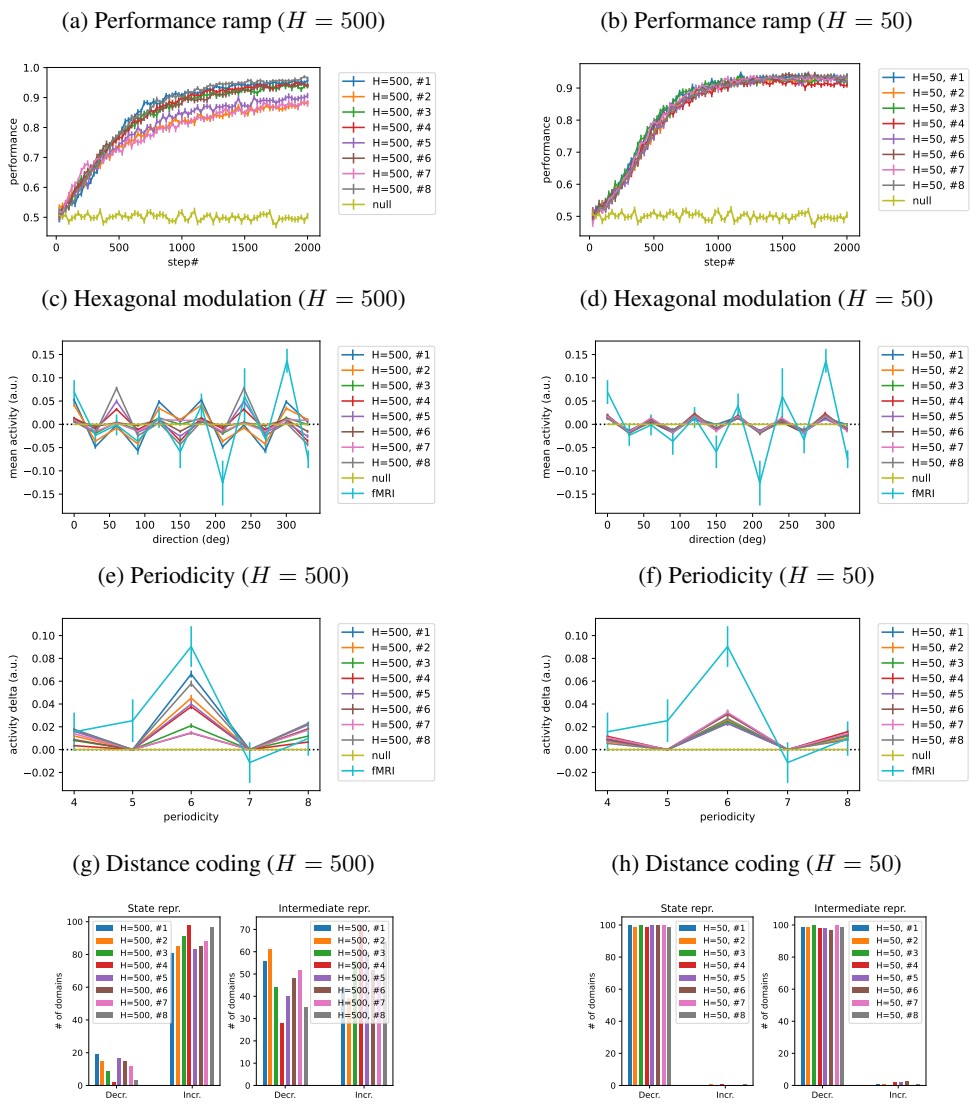

Figure 8: Results on the two-dimensional hierarchy task, analogous to Figure 3, from 8 ARDMOs with larger ($H = 500$) or smaller ($H = 50$) memory.

## D  LIMITATIONS

Currently, we know the following technical limitations of ARDMO.

- The model struggles in learning a larger scale of relational structures. For example, the performance notably drops for 20 entities or larger in the 1D case, and for $5 \times 5$ or larger in the 2D case.
- The model can handle the case with 3 or more relations (without axis hint like in our 2D case) but is limited to a small scale of relation structures. For example, in the case with 4 relations, learning becomes significantly slower for $4 \times 4$ entities and fails for larger cases.

## E  PLATFORM DETAILS

We used 3 computers with the following specifications: (1) 56 core CPUs (256G memory) with 4 V100 GPUs (16G memory each), (2) 56 core CPUs (256G memory) with 4 V100 GPUs (32G memory each), and (3) 96 core CPUs (256G memory) with 4 A100 GPUs (40G memory each). All code is implemented with Python (3.9.13) / Pytorch (1.12.1).

