# A Cognitive Model for Learning Abstract Relational Structures from Memory-based Decision-Making Tasks

## Abstract

Motivated by a recent neuroscientific hypothesis, some theoretical studies have accounted for the neural cognitive maps in the hippocampal formation as a representation of the general relational structure across task environments. However, despite their remarkable results reproducing place and grid cell properties, it is unclear whether their account can be extended to more general settings beyond spatial random-walk tasks in 2D environments. To address this question, we construct a novel cognitive model that performs memory-based decision-making tasks for learning abstract relational structures. Building on recent approaches of modular representation of abstract relations and concrete entities, we develop a learning algorithm that performs reward-guided relational inference across different entity domains, where we adopt a specific memory mechanism with content replacement to maintain dynamic binding between relations and entities. Our experiments show (i) the capability of our model to capture relational structures that can generalize over new domains with unseen entities, (ii) the difficulty of our task that leads existing powerful models, including Neural Turing Machine and vanilla Transformer, to complete failure, and (iii) the similarity of performance and internal representations of our model to recent human behavioral and fMRI experimental data, including distance coding and hexagonal modulation properties in the hippocampal formation.

## 1 Introduction

In everyday human cognition, we often find relationships among entities. Sometimes, we discern common relational structures across various domains and lift it to general knowledge (Figure 1). For example, ordering can be found not only among numbers, but also among objects and among individuals. In fact, ordering is so ubiquitous that it would be useful to consider the abstract notion of ordering irrespective of the concrete entities. This is just one example of *abstract relational structure*; other examples include tree-like structure, cyclic structure, and so on.

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

As a basic assessment, we ran the two trained models in the relational inference task in test domains and confirmed performance ramp to a reasonably high score for both models (Figure 5a). Our main interest here is, however, on how the learned intermediate ($h$) and state ($s$) representations compared to the human hippocampus and entorhinal cortex, respectively, in a similar task. For this, we performed the following analyses on the model after running it in each test domain.

---

[4]Note that both the model and human are much slower than the ideal observer (Section B.2.2), which reaches nearly 1.0 around 50 steps in our calculation.

[5]Since no other model so far has succeeded in solving our task and, we believe, neural plausibility of a non-working model is meaningless, we provide here no answer as to whether or not the similarity with the human fMRI data is unique to our model. In future studies, however, when other models turn out to be able to solve the task, the same question should be revisited and investigated.

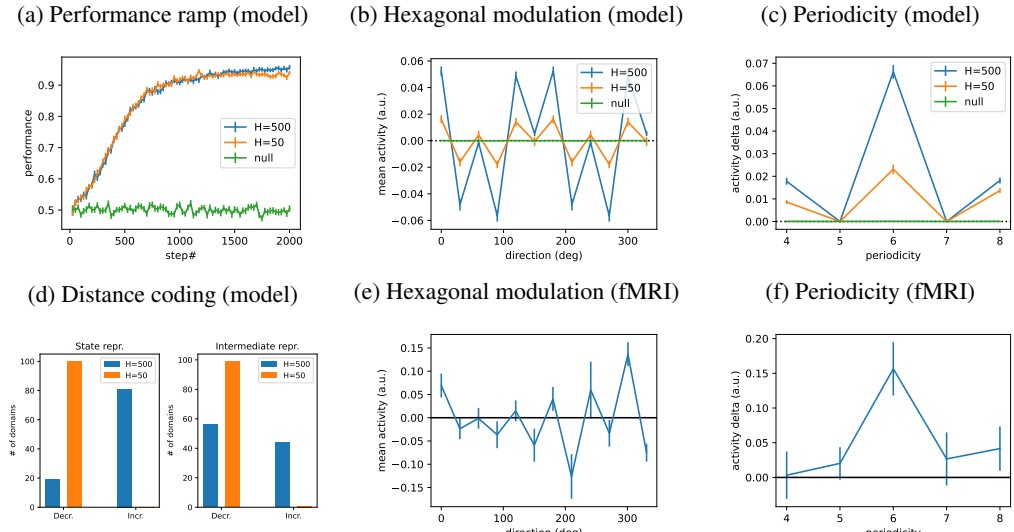

Figure 5: Results on the two-dimensional hierarchy task. (a) Performance ramp in the relational inference. (b, c) The hexagonal modulation property and estimated strength at each periodicity. (d) Distance coding in the state (left) or intermediate (right) representations. The results for models with larger ($H = 500$) or smaller memory ($H = 50$) as well as a null model are shown. (e, f) Analogous results to (b, c) in the fMRI experiment replotted from Park et al. (2021) (Fig. 4b). Error bars show standard errors over domains (models) or participants (humans).

First, Park et al. (2021) reported that fMRI signals from the entorhinal cortex during presentation of two consecutive stimuli showed hexagonal modulation with respect to the grid direction (Figure 6). To analyze our state representation in a compatible manner (Appendix C.1), we computed the (z-scored) state vectors corresponding to the images at each pair of grid positions and simply took their average. The state values, when plotted against the direction (angle) between the two positions, overall showed a periodic pattern of increase and decrease in the cycle of $60 \deg$; Figure 5b shows the phase-aligned mean within each bin of $30 \deg$. To quantify the periodicity, we calculated the mean differences between the state values at

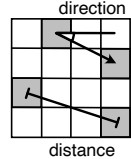

Figure 6: Grid coding

multiples of $60 \deg$ and the rest (6-fold periodicity) and repeated this for $90 \deg$, $72 \deg$, $51.4 \deg$, and $45 \deg$ (4-, 5-, 7-, and 8-fold periodicities, respectively); the 6-fold periodicity gave the maximal difference (Figure 5c). The results were overall similar between the two trained models (but with lower magnitudes for the smaller memory), resembling the human fMRI data (Park et al., 2021) (replotted in Figure 5ef).

Second, Park et al. (2021) observed that signals from the hippocampus and entorhinal cortex showed increasing patterns for distance coding (Figure 6). To analyze our state and intermediate representations in a compatible manner to the experiment, we computed the dissimilarity (Euclidean distance) between the vectors corresponding to the images at each pair of grid positions (Appendix C.1). Repeating these for different domains, the dissimilarity had either increasing or decreasing tendencies against the (Euclidean) distance in the grid space. (In other words, increasing tendency indicates that the representations are more similar between closer grid positions.) Figure 5d summarizes the number of domains with increase or decrease in the intermediate or state representation. In both representations, increasing tendencies were rarely found in the model with smaller memory, but much more often with larger memory, thus more similar to the human fMRI data reporting only increasing tendencies (Park et al., 2021). These results are robust across model instances trained in the same condition (Appendix C.2).

The hexagonal modulation in the human entorhinal cortex is closely related to the grid cell property found in the rodent entorhinal cortex (Hafting et al., 2005), since such modulation is expected as mixed neural signals from grid cells if they have hexagonal grid fields as in the rodent. In this sense,

our result reproducing the hexagonal modulation is related to previous model studies reproducing the grid cell property such as Whittington et al. (2018; 2020). However, we stress that neither result is directly predictable from the other since the task and model structures are rather different.

## 5 RELATED WORK

Recently, learning models for the cognitive map representations in the hippocampal formation have been drawing attention. Among others, Whittington et al. (2018; 2020) proposed TEM, which accounts for place and grid cells in rodents as an abstract relational structure of 2D-geometric environments while performing spatial random-walking tasks. Our study has been much influenced by theirs, adopting a similar modular architecture with separate relation and memory representations. However, the different task goal, decision-making, necessitates our model construction to use different approaches in (1) the learning algorithm that maximizes rewards from memory-based relational inference and (2) the memory mechanism with the update rule that can replace old contents. In particular, we showed that the Hopfield-type memory proposed by Ba et al. (2016) does not work in our setting (Section 4.1), which is quite striking, given that TEM was successful by using this mechanism: a simplistic adaption of TEM to our task fails. Later, Whittington et al. (2022) simplified TEM and clarified the relationship with Transformers (Vaswani et al., 2017). In this, they introduced another memory mechanism with a query-key-value-type read operation, but their update rule only adds new associations without erasing old ones (their Appendix). The modular architecture for learning to represent the general 2D structure has also been proposed by Uria et al. (2022), which also explained the variety of hippocampal cells. In different approaches, place and grid cell properties have also been reproduced in intermediate representations in recurrent networks (Banino et al., 2018; Sorscher et al., 2019) or successor representations in reinforcement learning (Stachenfeld et al., 2017).

Memory-augmented recurrent models have been used for solving general, complex tasks requiring relational reasoning. Some such models use memory for connecting abstract relations with concrete entities to discover abstract rules in input sequences and thereby solve symbol-processing tasks (Webb et al., 2020; Chen et al., 2021). However, despite the apparent similarity, their tasks to find out common rules hidden within fixed-length sequences are not compatible with our tasks to find out general structure in a set of binary relations. Graves et al. (2014; 2016) have proposed recurrent networks with external memory, highly influenced by von Neumann machines, that can learn to solve list and graph problems. Although these models are powerful, their way of coupling the recurrent network and the memory module seems to deteriorate disentangled representation of abstract relations and concrete domains (Section 4.1). Santoro et al. (2018) has presented a different approach for relational reasoning by using multi-head attention that allows for interaction between memory slots.

Recent studies have incorporated memory mechanisms in classifier neural network (Ba et al., 2016), in reinforcement learning (Pritzel et al., 2017; Hansen et al., 2018; Fortunato et al., 2019), in generative models (Li et al., 2016; Bornschein et al., 2017; Wu et al., 2018), in meta-learning (Santoro et al., 2016; Munkhdalai and Yu, 2017), and so on. Although these models have some technical commonalities with ours, they use memory to efficiently recall past experience and thereby increase specific task performance; therefore the goals are largely different from ours.

## 6 CONCLUSION

To pursue the recently emerging learning principle of abstract relational structure hypothesized for the hippocampal formation, we have developed a novel memory-based cognitive model for solving decision-making tasks and compared it with previous human behavioral and fMRI data. The results showing a good match suggest that this principle can potentially be extended from previously shown spatial random-walking to more general relational learning tasks. Future studies should further investigate neural representations of other relational structures like trees and cycles, interaction with other brain functions like the prefrontal cortex, and connection with neural data in rodents.

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

improves), and thereafter falls into another plateau. This stairway-like course continues for a few times and finally reaches a very long and slow slope for convergence. In some cases, the initial plateau is so long that it is unclear whether the optimization simply fails, in which case we manually stop it after around 2000 epochs and start it over.

### B.2.2  BASELINE MODELS

We adopted the following baseline models.

**Long short-term memory (LSTM) (Hochreiter and Schmidhuber, 1997)** A well-known recurrent model . Number of layers: 1. Hidden state dimensions: 70.

Table 1: Summary of results of relational inference (performance) and transitive inference (inference score) on one-dimensional hierarchy (at 1000 steps in each epoch). The mean and s.d. of scores over 8 model instances are shown.

|  | Ours | Ours w/o reg. | LSTM | TF | FW | Human (d1) | Human (d2) |
|---|---|---|---|---|---|---|---|
| Relational Inf. | $1.00 \pm 0.00$ | $1.00 \pm 0.00$ | $0.51 \pm 0.02$ | $0.50 \pm 0.00$ | $0.51 \pm 0.01$ | 0.96 | 0.97 |
| Transitive Inf. | $2.86 \pm 0.07$ | $2.09 \pm 0.14$ | — | — | — | 2.95 | 2.87 |

**Neural Turing Machine (NTM) (Graves et al., 2014)** A recurrent model with updatable external memory. Hidden state dimension: 20. Memory height (number of slots): 50. Memory width (content dimensions): 100

**Differentiable Neural Computer (DNC) (Graves et al., 2016)** An extension of NTM. Hidden state dimension: 20. Memory height: 50. Memory width: 100. Number of write heads: 1. Number of read heads: 4.

**vanilla Transformer (Vaswani et al., 2017)** A well-known language model incorporating self-attention with (additive) sine/cosine position-encoding and temporal masking. Number of layers: 2 (repeated). Number of heads: 1. Hidden state dimension: 1400. Drop-out rate: 0.1. Context size: 100.

**Fast Weight** A modified version of our model using the auto-associative memory proposed in Ba et al. (2016). The encoding/decoding method of state and content described in Whittington et al. (2020, Section 4) is used.

In all of these, we input the concatenation of two given images to the model at each step and the correct relation as target. For NTM and DNC, we also append the correct relation in one step to the input in the next step, so that the model can take the correctness of the previous inference into account. For Transformer, we give the concatenated image pair and the target alternately in the input and let the model infer the target from each image pair, whose summed squired error becomes the loss function.

In addition, we examined the ideal observer, which performs as well as possible in a given epoch, assuming that it knows perfectly the 1D task structure. We implemented the ideal observer such that it memorizes all past pairs in the epoch and, at each step, simply searches for the given pair or its flipped pair; if it finds, then it can give the right answer (reward 1) or otherwise a random answer (reward 0.5).

### B.3 ADDITIONAL RESULTS

Table 1 summarizes the results from 8 model instances for each task and for each model (except NTM and DNC for which we could train only one instance due to the very long training time), together with the human data. Figure 7 shows the traces of performance in relational inference and inference score in transitive inference for all models.

For Transformer, we tried a number of other architecture parameter settings than described in Section B.2.2, using more layers (5), more heads (2 or 5), lower input dimensions (2, 10, or, 100), or larger context sizes (300). (The hidden state dimensions were the same as the input dimensions.) The result was similar: all models completely failed even in training. This result was somewhat surprising to us since the task initially appeared to be easy for Transformer. Indeed, one can imagine a Transformer that uses the input directly as a query and key and then uses position encoding to find the corresponding relation found in the next entry in the sequence. However, optimization never found out such solution but seemed to just fall into meaningless local minima.

## C ADDENDUM ON THE EXPERIMENT IN SECTION 4.2

### C.1 ANALYSIS METHOD

Our aim here is to analyze the internal representations in out trained model in a way comparable to the brain analysis described in Park et al. (2021).

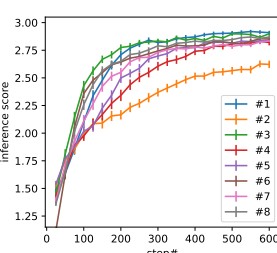

Figure 7: Additional results on one-dimensional hierarchy: (a) the traces of performance in relational inference and (b) inference score in transitive inference for all models.

First, in the fMRI experiment (Park et al., 2021), they analyzed the hexagonal modulation property of the brain signals from the entorhinal cortex during two consecutive visual stimuli corresponding to two grid positions. To simulate this, after running the model in a given test domain, we obtained the state vector corresponding to the image at each grid position by using inference through the memory and took the average over the two state vectors. The values in state values are normalized as z-scores. By plotting the state values in each dimension against the angle between the grid positions, we obtained a function over angles. Repeating this for different dimensions, we obtained a set of such functions. To see if those functions had periodicity in a given cycle ($90 \deg$, $72 \deg$, $60 \deg$, $51.4 \deg$, and $45 \deg$), we visualize the average over the functions, where we needed to deal with that each function may have a different phase. In Park et al. (2021), to avoid contamination of phase and periodicity, they estimated the phases by using another dataset obtained from a separate experiment with the same task condition. We simulated this by using the state vectors obtained through the model at 25 steps prior to the completion of running in the test domain (which has a different memory matrix). To estimate the phase, following the method in Park et al. (2021), we fit state values in each dimension against the sin and cos of the angle and took the arc-tangent of the regression coefficients. We then calculated the circular mean over the obtained phases. Finally, we took the phase-aligned average of the functions for all dimensions separately for each domain, and then averaged the resulting functions for all domains.

Second, in the same experiment Park et al. (2021), they analyzed the distance coding in the brain signals from the hippocampus and entorhinal cortex. To simulate this, we similarly obtained the intermediate or state vectors corresponding to the images at each pair of grid positions and calculated the distance between these as their dissimilarity. Although (Park et al., 2021) used Maharanobis distance, we used Euclidean distance since our case uses the deterministic state inference. By plotting the dissimilarity against the Euclidean distance between the grid positions, we determined the increase or decrease tendency by the correlation coefficient (positive or negative).

## C.2 ADDITIONAL RESULTS

We trained 4 models with larger ($H = 500$) or smaller ($H = 50$) memory. Figure 8 summarizes the results for these in the same format as Figure 5. These show overall robustness of the results across model instances.

## D PLATFORM DETAILS

We used 3 computers with the following specifications: (1) 56 core CPUs (256G memory) with 4 V100 GPUs (16G memory each), (2) 56 core CPUs (256G memory) with 4 V100 GPUs (32G memory each), and (3) 96 core CPUs (256G memory) with 4 A100 GPUs (40G memory each). All code is implemented with Python (3.9.13) / Pytorch (1.12.1).

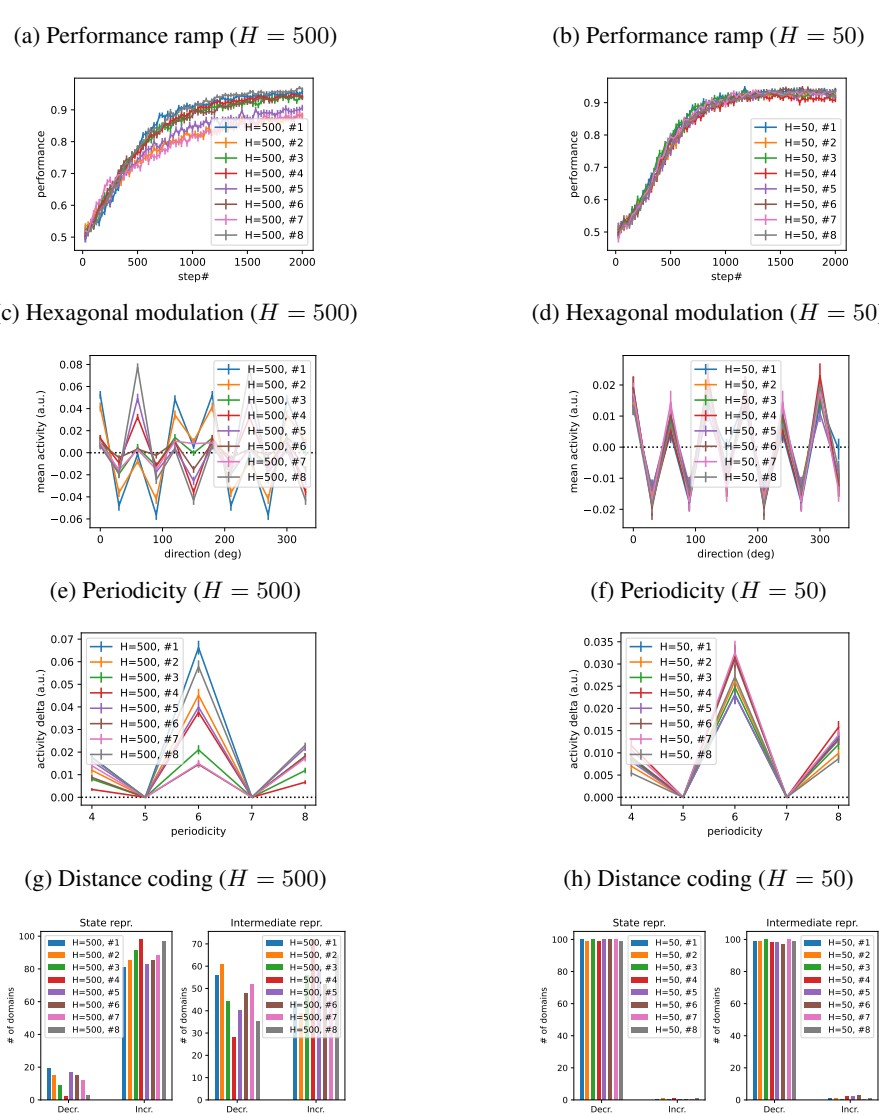

Figure 8: Results on the two-dimensional hierarchy task, analogous to Figure 5, from 8 models with larger ($H = 500$) or smaller ($H = 50$) memory.