# OpenReview forum: "A Cognitive Model for Learning Abstract Relational Structures from Memory-based Decision-Making Tasks"
_ICLR.cc/2024/Conference — ICLR 2024 poster_

### Official Review · Reviewer_NM51 · 2023-10-31

**Soundness:** 4 excellent
**Presentation:** 3 good
**Contribution:** 2 fair
**Rating:** 6
**Confidence:** 3

**Summary:**

This work builds a model that learns generalizable abstract relational structure on a decision making task where one has to answer the relation between pairs of stimuli for a reward. The stimuli can have "many to many" relations in that each stimuli can have up to 4 relations with other stimuli. The model builds on works like TEM and NTM where an explicit external memory module is written to/read from. The authors tried a one-dimensional version of the task where stimuli only had two relations and a two-dimensional version of the task where there can be more relations. The authors then saw that the model reproduced some neural phenomenon on such relational tasks such as distance coding, hexagonal modulation, and distance coding.

**Strengths:**

* Very comprehensive review of past work.

* Experiments are rigorous (multiple seeds, etc) and well-done. Showing the reproduction of the neural phenomenon is pretty nice.

* Model is written clearly.

**Weaknesses:**

* There is extensive discussion of previous work, but I was left wondering what exact contributions this model makes over other models in this space like TEM. The paper discusses numerous differences, but I would like to see explicit discussion of what this model brings to the table, what specific phenomenon that this model produces that other models don't, etc. There are maybe some signs of this throughout the paper, but I didn't see any explicit discussion on it.

* There's no limitations section/paragraph, which is an important part of any iclr paper.

**Questions:**

* Would it be possible for the model to learn these relations implicitly without specifically being rewarded for them? Sometimes for humans, abstract relational structure can often be learned in service of doing a specific task, rather than being trained specifically on finding the correct relations.

---

> ### Author Response · Authors · 2023-11-21
> **Response to Reviewer NM51**
>
> Thanks for the positive review.
>
> We think the exact contribution in TEM space as follows.  In TEM, they proposed the learning principle of abstract relational structure as hypothesis for hippocampal computation and a particular model to perform a random-walking task in this principle that can reproduce rodent data.  Our exact contribution is to consider another model in this principle that performs a decision making and reproduce human data, thus strengthening the hypothesis.   The paper explains that this is actually a substantial undertaking requiring a number of technical developments and experimental efforts.
>
> We miss the limitation section and will have it in the revision.  Right now, we think of limitation in the scalability of the relational structure.  The model finds difficulty structure larger than ~15 entities in 1D or ~5x5 entities in 2D.  Also, the long training time is a big issue.
>
> The question on the implicit learning is extreme interesting.  We have not thought about this possibility, and our current model would not be capable without substantial extension.  Probably, the task is a lot more difficult (note that the current task is quite difficult).

---

> > ### Comment · Reviewer_NM51 · 2023-11-21
> >
> > Thanks for the response. I will keep my score as is.

---

### Official Review · Reviewer_k38n · 2023-11-01

**Soundness:** 3 good
**Presentation:** 4 excellent
**Contribution:** 3 good
**Rating:** 8
**Confidence:** 3

**Summary:**

This paper presents a computational model for learning continuous spaces.  The primitives of the model include relationships, a, as well as entities x (just vectors in R^N).  To learn a one-dimensional space, the model is given two relations (analogous to greater than and lesser than); to learn a 2-dimensional space the model is given four relations (analogous to left vs right and above vs below).  The model's given a set of training data using near neighbors and then asked to generalize to pairs that were not observed (transitive inference).  And after learning a set of relationships, the model can generalize in that after learning a 1-D relationship among one set of stimuli, the model can more rapidly learn the analogous relationship among a second set of stimuli.  Notably, other approaches that are widely used in computational neuroscience (Tolman Eichenbaum machine, neural Turing machine, LSTM etc) not only fail to show these properties but can't learn the problem in the first place.

**Strengths:**

As far as I'm aware this is a completely novel approach.

The construction of ``internal spaces'' is an absolutely fundamental in computational cognitive neuroscience.

The idea of separating entities from relationships could be on the right track.

**Weaknesses:**

I wonder if it's possible to get TEM/transformer/NTM/etc to do something like this task if it's presented differently.

I found the connection to neuroscience very indirect, notwithstanding the observation that the similarity of internal states exhibits a distance effect and there's evidence for 60 degree symmetry.  This model is very abstract.

**Questions:**

The observation that transformers (for instance) don't learn these tasks is interesting.  Presumably, though this model is ill-suited for, say, language modeling.  What other problems can this computational approach solve (preferably in the general field of AI/ML)?

How does this approach scale?  As the number of continuous dimensions goes up how does it behave?  Suppose you chose a different way to tile the plane.  Rather than placing items at grid coordinates, suppose each item had N near neighbors (or that the exemplars were irregularly scattered).  This would mean that the number of relations a has to grow.  How sensitive is this model to the number of relations (controlling for the dimension of the space)?  Does it depend on a regular tiling of the space with entities?

An alternative approach is to simply assume that the brain is organized to represent low dimensional spaces and the learning problem is to determine how to map those internal spaces onto the external world. E.g.,
https://doi.org/10.1109/IJCNN54540.2023.10190998
https://doi.org/10.1109/IJCNN54540.2023.10191578

---

> ### Author Response · Authors · 2023-11-21
> **Response to Reviewer k38n**
>
> Thanks for the positive and useful comments.
>
> Regarding what other problems our model can solve, we consider that it can learn general tree or cyclic structures beyond the presented 1D/2D structures.  However, since our model focuses on the particular type of tasks of discovering abstract relation structure, it would need a substantial extension to solve a more general problem as in AI/ML.  We believe that this situation is not so unreasonable as this seems the usual tension between AI models and cognitive (or computational neuroscience) models.  AI models are usually designed to be general enough for solving various, interesting tasks with high performance, while they often have components unrelated to the task in question, thus can sometimes obscure how the models solve the task or fail.  On the other hand, cognitive models are often designed to understand certain cognitive or neural data at the abstract level and thus specialized to a particular setting and drop unnecessary components for the task in question, which often allow us to focus on, explain, and clarify the computational essence (Occam’s razor).  Neither approach is better than the other in general.  However, in our study, since our goal is to understand the novel kind of task (abstract relational learning in decision-making setting) and the related neural properties, we chose to design a cognitive model.
>
> Regarding scalability, in our experience, our model works for a range of state dimensions, 10 to 100 (in 1D case).  Interestingly, we observed that it stopped working for a larger dimension possibly because it could not find an optimal solution which allows for generalization over domains.  We will provide a plot in the revision.  As for the suggested different approach of tiling, which we believe is about general graph structure, we have not yet studied this direction and probably this would be the next step; thus, we do not have a clear answer to the question at this moment.  However, we are quite confident that the hexagonal modulation property is specific to the regular 2D tiling.
>
> Finally, thanks for the suggested references.  We will include and discuss these in the revision.

---

> > ### Comment · Reviewer_k38n · 2023-11-23
> > **response**
> >
> > I have read the author's response.
> >
> > These are interesting questions for future work.

---

### Official Review · Reviewer_rY1U · 2023-11-03

**Soundness:** 4 excellent
**Presentation:** 3 good
**Contribution:** 3 good
**Rating:** 8
**Confidence:** 4

**Summary:**

This paper describes a new neural model that learns abstract relations for one-dimensional and two-dimensional set orderings. The model is trained and tested on sets from different domains, such that  the testing phase is identical to human experiments on learning transitive relations.
The model is demonstrated to generalize to unseen domains, while several stat-of-the art methods are not able to solve their task. This generalization is made possible by the model's architecture with two components - (1) a set matrixes for learning abstract relations (one per relation), and set of  memory matrixes for binding concrete tokens from a specific domain to the abstract relations. The model is cognitively plausible, as its performance during the test phase appears to be similar to human performance in relation learning experiments, while other state-of-the-art models fail to complete the task.

**Strengths:**

1. This paper makes a novel contribution, significantly improving on existing models.

2. Model testing based on simulating previous human studies strongly supports cognitive plausibility of the model.

**Weaknesses:**

I found the presentation to be poorly readable in places - this is not a big deal, but I would suggest editing for clarity.

**Questions:**

The section analyzing hexagonal modulation within the model was not entirely clear to me -- it wasn't clear why the authors used the specific method of averaging state vectors.  Is there a citation, or maybe some explanation rationalizing this method? This can be included in the supplement. What is the significance of this hexagonal modulation emerging, given that the model's architecture is fundamentaly different from biological brains? Why would the authors expect hexagonal modulation to emerge?

---

> ### Author Response · Authors · 2023-11-21
> **Response to Reviewer rY1U**
>
> Thanks for the positive review.  We will certainly make the best efforts to improve the clarity especially in Section 4.2.
>
> Our approach of averaging state vectors comes from the analysis method used in Park et al. (2021).  In their experiment, they measured the BOLD activities while they consecutively presented two stimuli (corresponding to two positions in the map), and they took the average over all the activities at each voxel.  We simulated this by averaging the hidden variables corresponding to the two inputs.  This may sound a bit crude, but we believe this is the best we could do.  We have more details about this part in Sec C.1.
>
> On the significance of the emerging hexagonal modulation, this result suggests that the hippocampal formation may use the learning principle of abstract relational structure.  The model is certainly abstract and structurally different from the biological brain.  The question as to how these can correspond to each other is left open.  There could a neural implementation of our algorithm, or there could be some other model of the same principle for which a neural implementation could be found.  In any case, it is important to understand, from theoretical neuroscience point of view, why brain might have the hexagonal modulation property.  Pursing the principle is one such approach.
>
> The reason why we expected the emergence of hexagonal modulation is as follows.  Since the previous work by Whittington et al. (2018,2020) showed that hexagonal grid-cell property emerges from the abstract relation structure principle in a 2D random-walking task and since hexagonal modulation is supposed to arise as aggregate responses from grid-cells, we expected this property to emerge from the same principle in a different, decision-making task.  This is actually not so straightforward since there is a subtlety in the difference of the tasks.  In the random-walking task, the relations are all one-to-one and have no ambiguity, while in the decision-making task, the relations are many-to-many (Sec 2).  Nonetheless, the hexagonal representation appears to be optimal for 2D structure.

---

### Official Review · Reviewer_RHcS · 2023-11-03

**Soundness:** 2 fair
**Presentation:** 2 fair
**Contribution:** 3 good
**Rating:** 5
**Confidence:** 4

**Summary:**

This paper proposes a new cognitive model for performing memory-based decision-making tasks. The main contribution is a learning algorithm that allows the model to learn abstract relationships from reward-guided relational inference tasks, while maintaining dynamic binding between these abstract relations and concrete entities in a given task using a memory mechanism. The experiments demonstrate the model's ability to capture relational structures in one-dimensional and two-dimensional hierarchies. The authors also show that the model exhibits both performance and internal representations that bear resemblance to human behavioral and fMRI experimental data.

**Strengths:**

This paper introduces an interesting cognitive model for acquiring abstract relationships through reward-guided relational inference tasks. The experiments showcase the model's ability in learning relational structures that exhibit generalization across novel domains, featuring previously unseen entities. Notably, it significantly outperforms baseline models, such as LSTMs and standard Transformers. Further, the authors reveal an intriguing alignment between the model's behavior and fMRI data from humans performing the same tasks.

**Weaknesses:**

While the overall presentation of the paper is good, there are a couple of sections that are not easy to follow. The results on the two-dimensional hierarchy (section 4.2) can be challenging to understand for someone not very familiar with the findings in Park et al. (2021). Additionally, the notation in the section on transitive inference (3.4) can be a bit confusing (please see questions below).

The paper also misses references to related works on models for cognitive maps [1, 2]. Notably, [2] provides a unifying explanation for multiple hippocampal observations, while [3] presents an interesting approach for the reuse of learned abstractions in the form of graph schemas. It would be helpful to discuss the relationship between the approach in this work and these previous works.

Minor: There are several grammatical errors and a few typos (e.g., 'Maharanobis' on page 15) scattered throughout the paper.


[1] George, D., et al., 2021. Clone-structured graph representations enable flexible learning and vicarious evaluation of cognitive maps. Nature communications, 12(1), p.2392.

[2] Raju, R.V., et al.., 2022. Space is a latent sequence: Structured sequence learning as a unified theory of representation in the hippocampus.

[3] Guntupalli, J.S., et al., 2023. Graph schemas as abstractions for transfer learning, inference, and planning. arXiv preprint arXiv:2302.07350.

**Questions:**

- What properties does the relation matrix possess, could you offer insights into them?
- Is there a separate MLP for each $g_a$? If yes, how do you ensure the probabilities sum to 1?
- In equation 10, does $m$ in $\psi_a^{m-1}$ correspond to the $m^{\rm th}$ power? Or is it the $m^{\rm th}$ iteration? If the latter, how is  $\psi_a^{m-1}$ updated?
- In the formula for the inference score in section 4.1, what is $c^{ti}$?
- How was the value $\alpha=0.7$ chosen?
- In the caption of Figure 4, you mention that the for NTM and DNC the horizontal axis is 50-times reduced for readability. Does this mean that they used 50-times more epochs?
- How were the hyperparameters selected for the baseline methods in Figure 4a?
- In Figure 4b, why is the performance, after approximately 150 steps, slightly worse after 8000 epochs compared to the performance after 1000 epochs?
- What is the effect of S (the length of the state vector) on the results?
- In section 4.2, you discuss the learned intermediate representation $h$. I'd like to clarify the definition, since there appear to be two distinct uses of $h$ in equations 3 and 4.
- Could you please clarify how the state values are computed in section 4.2?
- Are there any thoughts about how this approach can be extended to more general relational graphs or to scenarios with sparse rewards?

---

> ### Author Response · Authors · 2023-11-21
> **Response to Reviewer RHcS**
>
> Thanks very much for the useful comments.  Especially, we will certainly incorporate and discuss the suggested references, which provide a different approach to explanation of hippocampal cognitive maps with abstraction and generalization.
>
> Answers to questions:
>
> * Regarding the relation matrix, we generally interpret it as a kind of rotation in the state space since 2D visualization after dimension reduction often indicates so, although it is often not very precisely so.
> * The MLP ends with a softmax function (Sec B.2.1)
> * $\phi_a^m(s)$ means m-th iteration, namely, $\phi_a(\phi_a(...(s)..))$, where $\phi_a$ is recursively applied m-times.  There is no update in this computation since $\phi_a(s)$ is defined as $\kappa(\rho(R_a s))$ where all the parameters used there $(M, W, R_a)$ are fixed when it is performed.  Note that we perform transitive inference task for testing a learned model (the model does not learn the transitive inference task).
> * c^ti is the confidence value for the relation chosen by the model: $\max_a p^+(a|s,s')$ (Sec 3.4).
> * In our experience, the model behaves equally well when alpha is large, say, $\alpha \geq 0.5$ but badly when it is small otherwise.  We chose $\alpha=0.7$ arbitrarily.  We will provide a plot in the revision.
> * On the number of steps in NTM and DNC, these indeed spent 50 times more epochs than the other models, taking about 40 days for the task (Fig 4).  Since the plot (Fig 4a) is in retrospect a bit confusing, in the revision, we will show a log plot (in x-axis).
> * We set the hyperparameters of the base line methods as follows.  For LSTM, we chose its state dimension as the state dimension plus the memory dimension of our model.  We also played with other settings but the result was pretty much the same.  For NTM and DNC, we used the default parameters of the package.  Since these take huge amount of time for training, we did not try other settings.  For Transformer, we chose the state dimension as the state dimension plus the memory dimension of our model.  We extensively searched other hyperparameters so that at least training works, as described in Sec B.2.2.  For FW, we set the same hyperparameters as our model except for $\alpha=0.5$ (similar to TEM).  Sec B.2.2 gives more on this topic.
> * As for the slight performance decrease in Fig 4b, we do not know exactly the reason but it is likely some kind of overfitting.
> * On the effect of $S$, we observed that the training works for a range of it from 10 to 100.  We will provide a plot in the revision.
> * The intermediate representation $h$ mentioned in Sec 4.2 refers to the $h$ in eq (4).
> * The state $s$ metioned in Sec 4.2 for given stimulus x is computed by $s=infer_{M,W}(x)$ using eq (4).  Note that we first run the model in the test domain (Sec 4.2, par 3)
> * For extension, we think that the current framework can be used for general relational graphs like trees and cycles.  For sparse rewards, the current design assumes immediate rewards: the reward is associated with the choice made in the same step.  Therefore we would likely need a nontrivial extension possibly with value functions.  We have not studied either topic much so far.

---

### Meta-Review · Area_Chair_kN1M · 2023-12-05

**Metareview:**

This paper presents a neuroscience-inspired model of decision making that relies on a memory module plus, a for representing abstract relations, and a module for representing concrete entities. The model engages in reward-guided relational inference across different domains via binding between the abstract relations and concrete entities. The authors show that the model is capable of learning relations that abstract over different domains (better than some previous models) and matches humans in both behavior and fMRI data.

The reviewers were fairly positive about this paper. There were some critiques regarding clarity of writing and clarity on relation to previous work, but there were no major technical concerns, and post-rebuttal, the scores were 5,8,8,6, which is clearly above the acceptance threshold. Thus an accept decision was reached.

**Justification For Why Not Higher Score:**

Although this paper was received positively, it is my assessment that it is not a huge advance beyond the Tolman-Eichenbaum Machine (Whittington et al., 2020). It's an advance, to be clear, but it is conceptually very similar. Thus, I don't think it deserves a spotlight.

**Justification For Why Not Lower Score:**

The reviewers had no technical complaints, and were generally positive, so I don't think it would be appropriate to reject this paper.

---

### Decision · Program_Chairs · 2024-01-16

Accept (poster)